# An Adaptive Tangent Feature Perspective
# of Neural Networks

Daniel LeJeune[1],* Sina Alemohammad[2]
[1]Stanford University, [2]Rice University
daniel@dlej.net, sa86@rice.edu

In order to better understand feature learning in neural networks, we propose and study linear models in tangent feature space where the features are allowed to be transformed during training. We consider linear feature transformations, resulting in a joint optimization over parameters and transformations with a bilinear interpolation constraint. We show that this relaxed optimization problem has an equivalent linearly constrained optimization with structured regularization that encourages approximately low rank solutions. Specializing to structures arising in neural networks, we gain insights into how the features and thus the kernel function change, providing additional nuance to the phenomenon of kernel alignment when the target function is poorly represented by tangent features. We verify our theoretical observations in the kernel alignment of real neural networks.

## 1. Introduction

Tremendous research effort has been expended in developing and understanding neural networks [1–4]. In terms of development, this effort has been met with commensurate tremendous practical success, dominating the state of the art [5–7] and establishing a new normal of replacing intricately engineered solutions with the conceptually simpler approach of learning from data [8].

Paradoxically, this simple principle of learning has not made the theoretical understanding of the success of neural networks easier [3, 9]. Neural networks stand in stark contrast to the engineered solutions that they have replaced—instead of leveraging vast amounts of human expertise about a particular problem, for which theoretical guarantees can be specifically tailored, neural networks appear to be universal learning machines, adapting easily to a wide range of tasks given enough data. The tall task of the theoretician is to prove that neural networks efficiently learn essentially any function of interest, while being trained through an opaque non-convex learning process on data with often unknown and mathematically uncharacterizable structure.

One promising theoretical direction for understanding neural networks has been through linearizing networks using the neural tangent kernel (NTK) framework [10, 11]. Given an appropriate initialization, infinitely wide neural networks can be shown to have constant gradients throughout training, such that the function agrees with its first-order Taylor approximation and is therefore a linear model, where the (tangent) features are the gradients of the network at initialization. The NTK framework reduces the complexity of neural networks down to linear (kernel) regression, which is significantly better theoretically understood [9, 12–14]. However, real neural networks still outperform their NTK approximants [15], and the fundamental assumption of the NTK—that the gradients do not change during training—is typically not satisfied in theory or practice [16, 17].

In this work, we take a step towards understanding the effects of the change of the gradients and how these changes allow the features to adapt to data. Rather than considering neural networks directly, we consider a relaxation of the problem in which the tangent features are allowed to adapt to the problem alongside the regression coefficients. We ask and answer the following questions:

*If allowed independent choice of features and coefficients near initialization, what solution is preferred? And can we explain observed feature alignment phenomena in neural networks by this mechanism?*

---

*Corresponding author.

First Conference on Parsimony and Learning (CPAL 2024).

Our specific contributions are as follows.

- We introduce a framework of linear feature adaptivity enabling two complementary views of the same optimization problem: as regression using adaptive features, and equivalently as structured regression using fixed features.

- We show how restricting the adaptivity imposes specific regularizing structure on the solution, resulting in a group approximate low rank penalty on a neural network based model.

- We consider the resulting adapted kernel and provide new insights on the phenomenon of NTK alignment [18–21], specifically when the target function is poorly represented using the initial tangent features.

Our work is quite far from an exact characterization of real neural networks. Nevertheless, our framework extends the class of phenomena can be explained by linear models built on tangent features, and so we believe it to be a valuable contribution towards understanding neural networks.

**Related work.** The neural tangent kernel has been proposed and studied extensively in the fixed tangent feature regime [10, 11, 15, 22]. Recent works have studied the alignment of the tangent kernel with the target function: Baratin et al. [18] empirically demonstrate tangent kernel alignment, and Atanasov et al. [19] show that linear network tangent kernels align with the target function early during training. Seleznova and Kutyniok [20] demonstrate alignment under certain initialization scalings, and characterize kernel alignment and neural collapse under a block structure assumption on the NTK [21]. In contrast, by characterizing the adaptive feature learning problem instead of neural networks specifically, we are able to gain more nuanced insights about kernel alignment. More similarly to our work, Radhakrishnan et al. [23, 24] show that the weight matrices in neural networks align with the average outer product of gradients.

The idea of simultaneously learning features and fitting regression models has appeared in the literature in tuning kernel parameters [25], multiple kernel learning [26], and automatic relevance determination (ARD) [27], which has been shown to correspond to a sparsifying iteratively reweighted $\ell_1$ optimization [28]. Other areas in which joint factorized optimization results in structured models include matrix factorization [29] and adaptive dropout [30], which are equivalent to iteratively reweighted $\ell_2$ optimization. Our work provides generic results on optimization of matrix products with rotationally invariant penalties, which complements the existing literature.

All proofs can be found in Appendix A.

## 2. An adaptive feature framework

We first formalize our adaptive feature framework, which enables us to jointly consider feature learning and regression. Our formulation is motivated by highly complex overparameterized models such as neural networks with rich tangent feature spaces.

**Notation.** Given a vector $\mathbf{v_x} \in \mathbb{R}^P$ parameterized by another vector $\mathbf{x} \in \mathbb{R}^Q$, we use the "denominator" layout of the derivative such that $\nabla_\mathbf{x} \mathbf{v_x} = \partial \mathbf{v_x} / \partial \mathbf{x} \in \mathbb{R}^{P \times Q}$, and given a scalar $v_\mathbf{X} \in \mathbb{R}$ parameterized by a matrix $\mathbf{X} \in \mathbb{R}^{P \times Q}$, we orient $\nabla_\mathbf{X} v_\mathbf{X} \in \mathbb{R}^{Q \times P}$. The vectorization of a matrix $\mathbf{X} = [\mathbf{x}_1 \quad \dots \quad \mathbf{x}_Q] \in \mathbb{R}^{P \times Q}$ is the stacking of columns such that $\mathrm{vec}(\mathbf{X})^\mathsf{T} = [\mathbf{x}_1^\mathsf{T} \quad \dots \quad \mathbf{x}_Q^\mathsf{T}] \in \mathbb{R}^{PQ}$. For $\mathbf{x} \in \mathbb{R}^{\min\{P,Q\}}$, we denote by $\mathrm{diag}_{P \times Q}(\mathbf{x}) \in \mathbb{R}^{P \times Q}$ the (possibly non-square) matrix with $\mathbf{x}$ along the main diagonal, and we omit the subscript $P \times Q$ when $P = Q$. We denote the set $\{1, \dots, N\}$ by $[N]$. Given a vector $\mathbf{x} \in \mathbb{R}^P$, we denote by $[\mathbf{x}]_j$ for $j \in [P]$ the $j$-th coordinate of $\mathbf{x}$. Given a matrix $\mathbf{X} \in \mathbb{R}^{P \times Q}$, we let $\sigma_i(\mathbf{X})$ denote its $i$-th largest singular value and $[\mathbf{X}]_{:j}$ for $j \in [Q]$ its $j$-th column. The characteristic function $\chi_\mathcal{A}$ of a set $\mathcal{A}$ satisfies $\chi_\mathcal{A}(\mathbf{x}) = 0$ for $\mathbf{x} \in \mathcal{A}$ and $\chi_\mathcal{A}(\mathbf{x}) = \infty$ for $\mathbf{x} \notin \mathcal{A}$.

## 2.1. First-order expansion with average gradients

Consider a differentiably parameterized function $f_{\boldsymbol{\theta}}\colon \mathbb{R}^D \to \mathbb{R}^C$ with parameters $\boldsymbol{\theta} \in \mathbb{R}^P$, such as a neural network. Our goal is to fit this function to data $(\mathbf{x}_1, \mathbf{y}_1), \ldots, (\mathbf{x}_N, \mathbf{y}_N) \in \mathbb{R}^{D \times C}$ by solving

$$\underset{\boldsymbol{\theta} \in \mathbb{R}^P}{\text{minimize}} \ \sum_{i=1}^{N} \ell(\mathbf{y}_i, f_{\boldsymbol{\theta}}(\mathbf{x}_i)),$$

where $\ell\colon \mathbb{R}^C \times \mathbb{R}^C \to \mathbb{R}$ is some loss function. In order characterize the solution, we need to understand how $f_{\boldsymbol{\theta}}$ changes with $\boldsymbol{\theta}$. One way that we can understand $f_{\boldsymbol{\theta}}$ is through the fundamental theorem of calculus for line integrals. Letting $\boldsymbol{\theta}_0$ be some reference parameters, such as random initialization or pretrained parameters,

$$f_{\boldsymbol{\theta}}(\mathbf{x}) - f_{\boldsymbol{\theta}_0}(\mathbf{x}) = \int_{\boldsymbol{\theta}_0}^{\boldsymbol{\theta}} \nabla_{\boldsymbol{\theta}'} f_{\boldsymbol{\theta}'}(\mathbf{x}) d\boldsymbol{\theta}' = \underbrace{\left( \int_0^1 \nabla_{\boldsymbol{\theta}'} f_{\boldsymbol{\theta}'}(\mathbf{x}) \big|_{\boldsymbol{\theta}'=(1-t)\boldsymbol{\theta}_0+t\boldsymbol{\theta}} dt \right)}_{\triangleq \overline{\nabla f_{\boldsymbol{\theta}}}(\mathbf{x})} (\boldsymbol{\theta} - \boldsymbol{\theta}_0).$$

That is, any such model is a linear predictor using the average **tangent features** $\overline{\nabla f_{\boldsymbol{\theta}}}(\mathbf{x})$ and coefficients $\boldsymbol{\theta} - \boldsymbol{\theta}_0$. When $\boldsymbol{\theta} \approx \boldsymbol{\theta}_0$, we should expect that $\overline{\nabla f_{\boldsymbol{\theta}}}(\mathbf{x}) \approx \nabla_{\boldsymbol{\theta}_0} f_{\boldsymbol{\theta}_0}(\mathbf{x})$, which are the tangent features at initialization. In fact, this has been shown to hold for very wide neural networks in the "lazy training" regime even for $\boldsymbol{\theta}$ at the end of training, in which case the problem can be understood as kernel regression using the neural tangent kernel [10, 11]. However, outside of those special circumstances, it is likely that $\overline{\nabla f_{\boldsymbol{\theta}}}(\mathbf{x})$ will change with $\boldsymbol{\theta}$. In general, it is difficult to say anything further about how $f_{\boldsymbol{\theta}}$ should change with $\boldsymbol{\theta}$, or what properties the optimal $\boldsymbol{\theta}$ and $\overline{\nabla f_{\boldsymbol{\theta}}}(\mathbf{x})$ for a prediction problem should have, and so we instead consider a related relaxation.

## 2.2. Relaxation: Interpolation with factorized features

Since overparameterized models typically admit infinitely many solutions, we first need to specify which solution we with to study. Due to the growing literature on interpolating predictors [9, 31–34] including neural networks, which are universal function approximators given enough parameters [35], we consider the popular minimum $\ell_2$ deviation interpolating solution:

$$\underset{\boldsymbol{\theta} \in \mathbb{R}^P}{\text{minimize}} \ \|\boldsymbol{\theta} - \boldsymbol{\theta}_0\|_2 \ \text{s.t.} \ f_{\boldsymbol{\theta}}(\mathbf{x}_i) = \widehat{\mathbf{y}}_i \triangleq \underset{\widetilde{\mathbf{y}} \in \mathbb{R}^C}{\arg \min} \ \ell(\mathbf{y}_i, \widetilde{\mathbf{y}}) \ \forall i \in [N].$$

This choice aligns with the common practice of training by gradient descent until training error is equal to zero. In classification problems, the minimizers are typically infinite valued and never realized unless there is label noise, in which case we should let $\widehat{\mathbf{y}}_i \triangleq \arg \min_{\widetilde{\mathbf{y}} \in \mathbb{R}^C} \sum_{j\,:\,\mathbf{x}_j=\mathbf{x}_i} \ell(\mathbf{y}_j, \widetilde{\mathbf{y}})$.

We now relax the problem to enable tractable analysis of feature adaptivity.

Consider a particular value of $\boldsymbol{\theta}$. If $P > N$ and the tangent feature space at initialization is very rich, then the range of the tensor formed by stacking all of the $\nabla_{\boldsymbol{\theta}_0} f_{\boldsymbol{\theta}_0}(\mathbf{x}_i)$ is likely to span $\mathbb{R}^{C \times N}$. As a result, there exists a matrix $\mathbf{M}_{\boldsymbol{\theta}} \in \mathbb{R}^{P \times P}$ such that for all $\mathbf{x}_i$, $\overline{\nabla f_{\boldsymbol{\theta}}}(\mathbf{x}_i) = \nabla_{\boldsymbol{\theta}_0} f_{\boldsymbol{\theta}_0}(\mathbf{x}_i) \mathbf{M}_{\boldsymbol{\theta}}$. Going one step further, if $N$ is sufficiently large and the gradients change sufficiently slowly in $\mathbf{x}$, we would expect $\mathbf{M}_{\boldsymbol{\theta}}$ to exist such that the linear equality holds even for test points $\mathbf{x}$ not in the training data. This matrix $\mathbf{M}_{\boldsymbol{\theta}}$ of course depends complexly and intricately on the parameterization of $f_{\boldsymbol{\theta}}$ and its initial parameters $\boldsymbol{\theta}_0$, and is no simpler to understand than the average tangent features.

Our key insight is that in complex nonlinear models such as deep neural networks, there may be many degrees of freedom in the parameters $\boldsymbol{\theta}$ that can change the tangent features without changing the dimensions of $\boldsymbol{\theta}$ that interact with those features. In this way, we speculate that such complex models are able to optimize both the features and coefficients jointly to serve the prediction goal. Based on this insight, we relax $\mathbf{M}_{\boldsymbol{\theta}}$ to be independent of $\boldsymbol{\theta}$, instead now a new variable $\mathbf{M} \in \mathbb{R}^{P \times P}$ to be optimized. Since we have applied an $\ell_2$ deviation penalty on $\boldsymbol{\theta}$, we need to also require that $\mathbf{M}$ should also not deviate much from its initial value of $\mathbf{M} = \mathbf{I}_P$. Hence finally, for some appropriate regularizer $\Omega\colon \mathbb{R}^{P \times P} \to \mathbb{R}$, we have our relaxed optimization problem

$$\widehat{\mathbf{M}}, \widehat{\boldsymbol{\theta}} \triangleq \underset{\mathbf{M} \in \mathbb{R}^{P \times P}, \boldsymbol{\theta} \in \mathbb{R}^P}{\arg \min} \ \Omega(\mathbf{M}) + \|\boldsymbol{\theta} - \boldsymbol{\theta}_0\|_2^2 \ \text{s.t.} \ \widehat{\mathbf{y}}_i - f_{\boldsymbol{\theta}_0}(\mathbf{x}_i) = \nabla_{\boldsymbol{\theta}_0} f_{\boldsymbol{\theta}_0}(\mathbf{x}_i) \mathbf{M}(\boldsymbol{\theta} - \boldsymbol{\theta}_0) \ \forall i \in [N]. \quad (1)$$

# 3. Adaptive feature learning

The learning problem in eq. (1) is very similar to regularized linear interpolation, except that it has a bilinear interpolation constraint (linear individually in each $\mathbf{M}$ and $\boldsymbol{\theta}$) rather than a simple linear constraint. This joint optimization can be characterized instead by an **effective penalty** $\widetilde{\Omega} \colon \mathbb{R}^P \to \mathbb{R}$ applied to $\boldsymbol{\beta} = \mathbf{M}\boldsymbol{\theta}$, or an equivalent learning problem

$$\widehat{\boldsymbol{\beta}} = \underset{\boldsymbol{\beta} \in \mathbb{R}^P}{\arg\min}\ \widetilde{\Omega}(\boldsymbol{\beta})\ \text{ s.t. }\ \widehat{\mathbf{y}}_i - f_{\boldsymbol{\theta}_0}(\mathbf{x}_i) = \nabla_{\boldsymbol{\theta}_0} f_{\boldsymbol{\theta}_0}(\mathbf{x}_i)\boldsymbol{\beta}\ \forall\, i \in [N]. \tag{2}$$

The resulting model can then be written as $f_{\widehat{\boldsymbol{\theta}}}(\mathbf{x}) = f_{\boldsymbol{\theta}_0}(\mathbf{x}) + \nabla_{\boldsymbol{\theta}_0} f_{\boldsymbol{\theta}_0}(\mathbf{x})\widehat{\boldsymbol{\beta}}$, simply a linear model in the tangent features. By mapping $\Omega$ to $\widetilde{\Omega}$, we can understand the effect of joint feature and coefficient learning on the use of the initial tangent features in the final model. Unfortunately, $\widetilde{\Omega}$ does not have an interpretable form in general.

However, a very natural restriction to the choice of $\Omega$ results in $\widetilde{\Omega}$ that is straightforward to describe and interpret. A priori, we have no knowledge of which directions in feature space are most important, so we can encourage search in all directions equally by choosing $\Omega$ to be *rotationally invariant*. We consider regularizers built from the following class of spectral regularizers for a strictly quasi-convex function $\omega \colon \mathbb{R} \to \mathbb{R}$ having minimum value $\omega(1) = 0$:

$$\Omega_\omega(\mathbf{M}) = \sum_{j=1}^{P} \omega(\sigma_j(\mathbf{M})).$$

This penalty applies only to the singular values $\sigma_j$, so the singular vectors of $\mathbf{M}$ are free to be rotated arbitrarily. Note that this choice ensures a tendency towards the initial value of $\mathbf{M}_{\boldsymbol{\theta}_0} = \mathbf{I}_P$, and it also means that we can consider symmetric positive semidefinite $\mathbf{M}$ without loss of generality.[1] A simple example of a choice for $\omega$ is $\omega(v) = |v - 1|^p$ for $p > 0$.

Some feature transformations might be particularly unnatural, especially when there is structure in the parameterization of the function, such as weight matrices at different layers of a neural network. We can encode this structure by applying a penalty $\Omega$ to sub-blocks of $\mathbf{M}$ independently. As we shall see, the presence of such structure has a significant impact on the resulting effective optimization.

When considering solutions, we are also interested in the kernel corresponding to the learned features. Specifically, for $\mathbf{x}, \mathbf{x}' \in \mathbb{R}^D$ we define the **adapted kernel** as the feature inner products

$$K(\mathbf{x}, \mathbf{x}') \triangleq \overline{\nabla f_{\widehat{\boldsymbol{\theta}}}}(\mathbf{x})\overline{\nabla f_{\widehat{\boldsymbol{\theta}}}}(\mathbf{x}')^{\mathsf{T}} = \nabla_{\boldsymbol{\theta}_0} f_{\boldsymbol{\theta}_0}(\mathbf{x})\widehat{\mathbf{M}}^2 \nabla_{\boldsymbol{\theta}_0} f_{\boldsymbol{\theta}_0}(\mathbf{x}')^{\mathsf{T}} \in \mathbb{R}^{C \times C}.$$

We correspondingly define the initial kernel $K_0(\mathbf{x}, \mathbf{x}') = \nabla_{\boldsymbol{\theta}_0} f_{\boldsymbol{\theta}_0}(\mathbf{x})\nabla_{\boldsymbol{\theta}_0} f_{\boldsymbol{\theta}_0}(\mathbf{x}')^{\mathsf{T}}$ for $\mathbf{M} = \mathbf{I}_P$, which is equal to the standard neural tangent kernel in neural networks.

## 3.1. Structureless feature learning

It is instructive to first consider the effect of unrestricted optimization of $\mathbf{M}$; that is, if the features were allowed to change without any structural constraints on how features and parameters must interact. In this case, we simply solve eq. (1) directly, using $\Omega = \Omega_\omega$ applied to the full $\mathbf{M}$.

**Theorem 1.** *There is a solution to eq. (1) with $\Omega = \Omega_\omega$ satisfying*

$$\widehat{\mathbf{M}} = \mathbf{I}_P + (s-1)\|\widehat{\boldsymbol{\beta}}\|_2^{-2}\widehat{\boldsymbol{\beta}}\widehat{\boldsymbol{\beta}}^{\mathsf{T}} \quad \text{and} \quad \widehat{\boldsymbol{\theta}} = \boldsymbol{\theta}_0 + s^{-1}\widehat{\boldsymbol{\beta}}.$$

*where $s = \arg\min_{z \geq 1} \omega(z) + \frac{\|\widehat{\boldsymbol{\beta}}\|_2^2}{z^2}$ and $\widehat{\boldsymbol{\beta}}$ is given by eq. (2) with $\widetilde{\Omega} = \|\cdot\|_2$. Furthermore, the adapted kernel for this solution is given by*

$$K(\mathbf{x}, \mathbf{x}') = K_0(\mathbf{x}, \mathbf{x}') + (s^2 - 1)\|\widehat{\boldsymbol{\beta}}\|_2^{-2} \underbrace{(f_{\widehat{\boldsymbol{\theta}}}(\mathbf{x}) - f_{\boldsymbol{\theta}_0}(\mathbf{x}))(f_{\widehat{\boldsymbol{\theta}}}(\mathbf{x}') - f_{\boldsymbol{\theta}_0}(\mathbf{x}'))^{\mathsf{T}}}_{\triangleq K_{\widehat{\mathbf{y}}}(\mathbf{x}, \mathbf{x}')}.$$

---

[1]Let $\mathbf{U}\mathbf{S}\mathbf{V}^{\mathsf{T}}$ be the SVD of any $\mathbf{M}$. We can always consider instead $\mathbf{M}' = \mathbf{U}\mathbf{S}\mathbf{U}^{\mathsf{T}}$ and $\boldsymbol{\theta}' = \mathbf{U}\mathbf{V}^{\mathsf{T}}\boldsymbol{\theta}$ such that the penalty value remains the same, and $\boldsymbol{\beta} = \mathbf{M}'\boldsymbol{\theta}' = \mathbf{M}\boldsymbol{\theta}$ and thus the linear constraints to not change.

*Proof sketch.* By a straightforward argument, the leading singular vector of $\mathbf{M}$ must be aligned with $\boldsymbol{\theta} - \boldsymbol{\theta}_0$ to minimize $\|\boldsymbol{\theta} - \boldsymbol{\theta}_0\|_2^2$ given the singular values of $\mathbf{M}$. For the leading singular value $s$, the equivalent solution must satisfy $\|\boldsymbol{\beta}\|_2 = s\|\boldsymbol{\theta}\|_2$. We can obtain $s$ given $\|\boldsymbol{\beta}\|_2$ by performing the minimization of the penalty $\omega(s) + \|\boldsymbol{\theta}\|_2^2 = \omega(s) + \frac{\|\boldsymbol{\beta}\|_2^2}{s^2}$, which is an increasing function of $\|\boldsymbol{\beta}\|_2$. □

Surprisingly, since $\widetilde{\Omega} = \|\cdot\|_2$, the equivalent solution in $\widehat{\boldsymbol{\beta}}$ is exactly that of ridgeless regression [32] using the initial tangent kernel features $\nabla_{\boldsymbol{\theta}_0} f_{\boldsymbol{\theta}_0}(\mathbf{x})$—therefore, when no structure is imposed on the adaptive features, the resulting predictions are simply the same as in NTK regression. However, even though the predictions are no different, we already can see a qualitative difference in the description of the system from NTK analysis. Specifically, this adaptive feature perspective reveals how the adapted kernel itself changes: it is a low rank perturbation to the original kernel that directly captures the model output as the label kernel $K_{\widehat{\mathbf{y}}}$. This kernel alignment effect has been empirically observed in real neural networks that depart from the lazy-training regime [18–20].

Moreover, the strength of the kernel alignment is directly related to the difficulty of the regression task as measured by the norm of $\widehat{\boldsymbol{\beta}}$. Note that $s$ takes minimum value at $\|\widehat{\boldsymbol{\beta}}\|_2 = 0$ and is increasing otherwise, which means that if the initial tangent features $\nabla_{\boldsymbol{\theta}_0} f_{\boldsymbol{\theta}_0}(\mathbf{x})$ are sufficient to fit $\widehat{y}_i$ with a small $\widehat{\boldsymbol{\beta}}$, then $s \approx 1$ and $K(\mathbf{x}, \mathbf{x}') \approx K_0(\mathbf{x}, \mathbf{x}')$.[2] It is only when the task is difficult and a larger $\widehat{\boldsymbol{\beta}}$ is required that $s^2 - 1 \gg 0$ and we observe kernel alignment with the label kernel $K_{\widehat{\mathbf{y}}}$. We illustrate this in an experiment with real neural networks on an MNIST regression task in Figure 1.

In order to go beyond ridgeless regression with feature learning, it is necessary to further constrain the structure of $\mathbf{M}$. By restricting $\mathbf{M}$ to operate independently on separate subspaces, the result is an effective group sparse penalty over the subspaces. As an extreme example that we detail in Appendix A.1, if $\mathbf{M}$ is constrained to be diagonal, operating independently on $P$ orthogonal subspaces of dimension 1, then the result is that $\widetilde{\Omega}$ is a conventional sparsity-inducing penalty. Concretely, we have $\widetilde{\Omega} = \|\cdot\|_1$ in the special case of $\Omega(\mathbf{M}) = \|\mathbf{M}\|_F^2$ with a diagonal constraint.

## 3.2. Neural network structure

We now wish to specialize to structures that arise in neural networks. The developments in this section will be built around parameterization of the network by matrices, but these results could be extended straightforwardly to higher order tensors (such as filter kernels in convolutional neural networks). Generically, the parameters of a neural network are $\boldsymbol{\theta} = \text{concat}(\text{vec}(\mathbf{W}_1), \ldots, \text{vec}(\mathbf{W}_L))$, where $\mathbf{W}_\ell \in \mathbb{R}^{P_\ell \times Q_\ell}$ are parameter matrices such that $P = \sum_{\ell=1}^L P_\ell Q_\ell$. These matrices comprise addition and multiplication operations in a directed acyclic graph along with other operations such as nonlinearities and batch or layer normalizations, which do not have parameters and thus do not contribute to the number of tangent features.

We first state our neural network model, and then we explain the motivation of each component.

**Model A.** *The neural network model has the following properties:*

1. *The parameter vector $\boldsymbol{\theta} - \boldsymbol{\theta}_0 \in \mathbb{R}^P$ consists of $L$ matrices $\mathbf{W}_1 \in \mathbb{R}^{P_1 \times Q_1}, \ldots, \mathbf{W}_L \in \mathbb{R}^{P_L \times Q_L}$.*

2. *The feature transformation operator $\mathbf{M} \in \mathbb{R}^{P \times P}$ is parameterized by $\mathbf{M}_\ell^{(1)} \in \mathbb{R}^{P_\ell \times P_\ell}$ and $\mathbf{M}_\ell^{(2)} \in \mathbb{R}^{Q_\ell \times Q_\ell}$ for each $\ell \in [L]$ such that application of $\mathbf{M}$ to $\boldsymbol{\theta}$ results in the mapping $(\mathbf{W}_\ell)_{\ell=1}^L \mapsto (\mathbf{M}_\ell^{(1)} \mathbf{W}_\ell \mathbf{M}_\ell^{(2)})_{\ell=1}^L$.*

3. *For strictly quasi-convex functions $\omega_\ell^{(1)}$ and $\omega_\ell^{(2)}$ for each $\ell \in [L]$ minimized at $\omega_\ell^{(1)}(1) = 0$ and $\omega_\ell^{(2)}(1) = 0$, the regularizer is given by $\Omega(\mathbf{M}) = \sum_{\ell=1}^L \Omega_{\omega_\ell^{(1)}}(\mathbf{M}_\ell^{(1)}) + \Omega_{\omega_\ell^{(2)}}(\mathbf{M}_\ell^{(2)})$.*

---

[2]While there is a $\|\widehat{\boldsymbol{\beta}}\|_2^{-2}$ factor as well, this is always canceled by the implicit $\|\widehat{\boldsymbol{\beta}}\|_2^2$ in $K_{\widehat{\mathbf{y}}}$, such that the overall scale of $K_{\widehat{\mathbf{y}}}$ is determined entirely by the factor $(s^2 - 1)$.

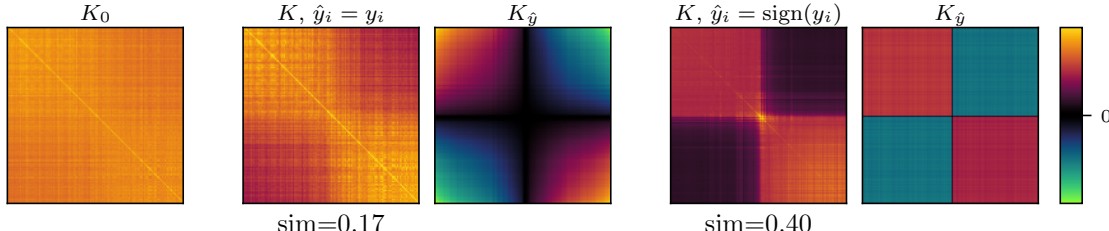

$$K_0 \qquad K, \hat{y}_i = y_i \qquad K_{\hat{y}} \qquad\qquad K, \hat{y}_i = \text{sign}(y_i) \qquad K_{\hat{y}}$$

sim=0.17 sim=0.40

Figure 1: **A more difficult task yield higher label kernel alignment.** We perform regression using a multi-layer perceptron on 500 MNIST digits from classes 2 and 3. We construct target labels $y_i$ as the best linear fit of binary $\pm 1$ labels using random neural network features. Then we train two networks, one (**left**) trained to predict $y_i$, and one (**right**) trained to predict $\text{sign}(y_i)$. We present the adapted kernel and label kernel matrices for data points ordered according to $y_i$ and report the cosine similarity of the adapted kernel and the label kernel. The harder task of regression with binarized labels has a higher label kernel alignment. Further details are given in Appendix B.1.

4. *The final matrix $\mathbf{W}_L$ has $Q_L = C$ and a fixed transformation of $\mathbf{M}_L^{(2)} = \mathbf{I}_C$ (corresponding to $\omega_L^{(2)} = \chi_{\{1\}}$), and there is a mapping $\mathbf{z} \colon \mathbb{R}^D \to \mathbb{R}^{P_\ell}$ such that $\nabla_{\text{vec}(\mathbf{W}_L)} f_{\boldsymbol{\theta}_0}(\mathbf{x}) = \mathbf{I}_C \otimes \mathbf{z}(\mathbf{x})^\mathsf{T}$.*

The first component is simply a reparameterization of $\mathbf{W}_\ell$ as the difference from initialization, to simplify notation. The other components are motivated as follows.

**Feature transformations.** Consider a single weight matrix $\mathbf{W}_\ell$ and the gradient of the $k$-th output $f_{\boldsymbol{\theta}}^{(k)}(\mathbf{x})$. The matrix structure of $\mathbf{W}_\ell$ limits the way in which the gradient tends to change. As such, rather than considering a $P_\ell Q_\ell \times P_\ell Q_\ell$ matrix $\mathbf{M}_\ell$ such that $\overline{\nabla_{\text{vec}(\mathbf{W})_\ell} f_{\boldsymbol{\theta}}}(\mathbf{x}) = \nabla_{\text{vec}(\mathbf{W}_\ell)} f_{\boldsymbol{\theta}_0}(\mathbf{x}) \mathbf{M}_\ell$, it is more natural to consider a Kronecker factorization $\mathbf{M}_\ell = \mathbf{M}_\ell^{(2)\mathsf{T}} \otimes \mathbf{M}_\ell^{(1)}$ such that

$$\overline{\nabla_{\mathbf{W}_\ell} f_{\boldsymbol{\theta}}}^{(k)}(\mathbf{x}) = \mathbf{M}_\ell^{(2)} \nabla_{\mathbf{W}_\ell} f_{\boldsymbol{\theta}_0}^{(k)}(\mathbf{x}) \mathbf{M}_\ell^{(1)},$$

which corresponds to the mapping $\mathbf{W}_\ell \mapsto \mathbf{M}_\ell^{(1)} \mathbf{W}_\ell \mathbf{M}_\ell^{(2)}$.

**Independent optimization.** We assume that the feature transformations $\mathbf{M}_\ell^{(1)}$ and $\mathbf{M}_\ell^{(2)}$ are independently optimized from each other and from the feature transformations corresponding to other weights $\ell' \neq \ell$. The motivation for this assumption comes from the fact that at initialization in wide fully connected neural networks, the gradients at each layer are known to be uncorrelated [36]. Because of independence within each layer, we also need to define the **joint penalty** for $v \geq 1$ as

$$(\omega_1 \oplus \omega_2)(v) \triangleq \min_{1 \leq z \leq v} \omega_1(z) + \omega_2(\tfrac{v}{z}).$$

It is straightforward to verify that if $\omega_1$ and $\omega_2$ are strictly quasi-convex such that $\omega_1(1) = \omega_2(1) = 0$, then $\omega_1 \oplus \omega_2$ has the same properties; see Appendix A.2. We also take this opportunity to define the scalar effective penalty and its induced effective penalty, which will simplify notation:

$$\tilde{\omega}(v) \triangleq \min_{z \geq 1} \omega(z) + \frac{v^2}{z^2} \quad \text{and} \quad \widetilde{\Omega}_\omega(\mathbf{M}) \triangleq \Omega_{\tilde{\omega}}(\mathbf{M}).$$

**Final weight matrix.** The final operation in most neural networks is linear matrix multiplication by the final weight matrix $\mathbf{W}_L \in \mathbb{R}^{P_L \times C}$, such that if $\mathbf{z}_{\boldsymbol{\theta}}(\mathbf{x}) \in \mathbb{R}^{P_L}$ are the penultimate layer features, then the output is a vector $f_{\boldsymbol{\theta}}(\mathbf{x}) = \mathbf{W}_L^\mathsf{T} \mathbf{z}_{\boldsymbol{\theta}}(\mathbf{x}) \in \mathbb{R}^C$. Noting that $\mathbf{z}_{\boldsymbol{\theta}}(\mathbf{x}) = \mathbf{M}_L^{(1)} \mathbf{z}_{\boldsymbol{\theta}_0}(\mathbf{x})$, we have the Kronecker formulation

$$f_{\boldsymbol{\theta}}(\mathbf{x}) = \underbrace{(\mathbf{I}_C \otimes (\mathbf{z}_{\boldsymbol{\theta}_0}(\mathbf{x})^\mathsf{T} \mathbf{M}_L^{(1)}))}_{\overline{\nabla_{\text{vec}(\mathbf{W}_L)} f_{\boldsymbol{\theta}}}(\mathbf{x})} \text{vec}(\mathbf{W}_L).$$

With the above neural network model in hand, this brings us to our main result on the solution to the adaptive feature learning problem.

**Theorem 2.** *There is a solution to eq. (1) under Model A such that for each $\ell \in [L]$,*

$$\widehat{\mathbf{M}}_\ell^{(1)} = \mathbf{U}_\ell \mathbf{S}_\ell^{(1)} \mathbf{U}_\ell^\mathsf{T}, \quad \widehat{\mathbf{W}}_\ell = \mathbf{U}_\ell \boldsymbol{\Sigma}_\ell \mathbf{V}_\ell^\mathsf{T}, \quad \widehat{\mathbf{M}}_\ell^{(2)} = \mathbf{V}_\ell \mathbf{S}_\ell^{(2)} \mathbf{V}_\ell^\mathsf{T},$$

*where $\mathbf{U}_\ell \in \mathbb{R}^{P_\ell \times P_\ell}$, $\mathbf{V}_\ell \in \mathbb{R}^{Q_\ell \times Q_\ell}$ are orthogonal matrices and $\mathbf{S}_\ell^{(1)} = \mathrm{diag}(\mathbf{s}_\ell^{(1)})$, $\boldsymbol{\Sigma}_\ell = \mathrm{diag}_{P_\ell \times Q_\ell}(\boldsymbol{\sigma}_\ell)$, $\mathbf{S}_\ell^{(2)} = \mathrm{diag}(\mathbf{s}_\ell^{(2)})$ are given by minimizers*

$$[\mathbf{s}_\ell^{(1)}]_j, [\boldsymbol{\sigma}_\ell]_j, [\mathbf{s}_\ell^{(2)}]_j = \operatorname*{arg\,min}_{s_1, s_2 \geq 1, \sigma \geq 0} \omega_\ell^{(1)}(s_1) + \omega_\ell^{(2)}(s_2) + \sigma^2 \ \text{ s.t. } \ s_1 \sigma s_2 = [\mathbf{d}_\ell]_j \ \text{ for } j \leq \min\{P_\ell, Q_\ell\}$$

*and $[\mathbf{s}_\ell^{(1)}]_j = 1$, $[\mathbf{s}_\ell^{(2)}]_j = 1$ for $j > \min\{P_\ell, Q_\ell\}$, such that $\widehat{\mathbf{B}}_\ell = \mathbf{U}_\ell \mathrm{diag}_{P_\ell \times Q_\ell}(\mathbf{d}_\ell) \mathbf{V}_\ell^\mathsf{T}$ satisfy*

$$(\widehat{\mathbf{B}}_\ell)_{\ell=1}^L \in \operatorname*{arg\,min}_{\mathbf{B}_\ell \in \mathbb{R}^{P_\ell \times Q_\ell}} \sum_{\ell=1}^L \widetilde{\Omega}_{\omega_\ell^{(1)} \oplus \omega_\ell^{(2)}}(\mathbf{B}_\ell) \ \text{ s.t. } \ \widehat{\mathbf{y}}_i - f_{\boldsymbol{\theta}_0}(\mathbf{x}_i) = \sum_{\ell=1}^L \nabla_{\mathrm{vec}(\mathbf{W}_\ell)} f_{\boldsymbol{\theta}_0}(\mathbf{x}_i) \mathrm{vec}(\mathbf{B}_\ell) \ \forall \ i \in [N].$$

*Furthermore, the adapted kernel for this solution is given by*

$$K(\mathbf{x}, \mathbf{x}') = K_0(\mathbf{x}, \mathbf{x}') + \sum_{\ell=1}^L \sum_{j=1}^{\min\{P_\ell, Q_\ell\}} ([\mathbf{s}_\ell^{(1)}]_j^2 [\mathbf{s}_\ell^{(2)}]_j^2 - 1) \nabla_{\mathrm{vec}(\mathbf{W}_\ell)} f_{\boldsymbol{\theta}_0}(\mathbf{x}) \mathrm{vec}([\mathbf{U}_\ell]_{:j} [\mathbf{V}_\ell]_{:j}^\mathsf{T})$$
$$\cdot \mathrm{vec}([\mathbf{U}_\ell]_{:j} [\mathbf{V}_\ell]_{:j}^\mathsf{T})^\mathsf{T} \nabla_{\mathrm{vec}(\mathbf{W}_\ell)} f_{\boldsymbol{\theta}_0}(\mathbf{x}')^\mathsf{T}.$$

The proof approach is similar to Theorem 1. In other words, we have reduced the bilinearly constrained optimization over $\mathbf{M}$ and $\boldsymbol{\theta}$ to a linearly constrained optimization over $(\mathbf{B}_\ell)_{\ell=1}^L$, just as we did in the unstructured case. However, this time, due to the matrix stucture of $\mathbf{W}_\ell$ and corresponding structure in $\mathbf{M}_\ell^{(1)}$ and $\mathbf{M}_\ell^{(2)}$, the new equivalent optimization is much richer than simple minimum $\ell_2$ norm interpolation using tangent features. To better understand the regularization in this new problem, we have the following result about the scalar effective penalty.

**Proposition 3.** *Let $\omega$ be a continuous strictly quasi-convex function minimized at $\omega(1) = 0$. Then $v^2 \mapsto \tilde{w}(v)$ is an increasing concave function, and $\tilde{\omega}(v) = v^2 + o(v^2)$ as $v \to 0$.*

That is, no matter the original penalties $\omega_\ell^{(1)}$ and $\omega_\ell^{(2)}$, the resulting $\widetilde{\Omega}_{\omega_\ell^{(1)} \oplus \omega_\ell^{(2)}}$ will always be a spectral penalty with 1) sub-quadratic tail behavior, and 2) quadratic behavior for small values. We illustrate this for a few examples in Figure 2. For one example, when $\omega(v) = (v - 1)^2$, the effective penalty $\tilde{\omega}(v)$ behaves like $|v|$ for large $v$, making $\widetilde{\Omega}_\omega(\mathbf{B})$ like the nuclear norm for large singular values and like the Frobenius norm for small singular values. In general, $\tilde{\omega}$ has slower tails than $\omega$. This behavior is highly related of the equivalence of the nuclear norm as the sum of Frobenius norms of two factors [29], which coincides with the case $\omega(v) = v^2$; however, since we constrain the $\mathbf{M}$ to be near $\mathbf{I}_P$, we retain Frobenius norm behavior near 0. The sub-quadratic nature of $\tilde{\omega}$ is a straightforward consequence of Legendre–Fenchel conjugacy.

The result of this effective regularization is a model that is able to leverage structures through the group approximate low-rank penalty while also being robust to noise and model misspecification through the Frobenius norm penalty for small singular values. From this perspective, we can conceptually consider a decomposition of the matrices $\mathbf{B}_\ell \approx \mathbf{B}_\ell^{\mathrm{LO}} + \mathbf{B}_\ell^{\mathrm{NTK}}$, where $\mathbf{B}_\ell^{\mathrm{LO}}$ are low rank and capture the structure learned from the data that is predictive of the training labels, while $\mathbf{B}_\ell^{\mathrm{NTK}}$ form the component that fits the residual (after regression using $\mathbf{B}_\ell^{\mathrm{LO}}$) via standard NTK interpolation with Frobenius norm minimization. In this way, the adaptiveness of the neural network is able to get the benefits of both strategies.

To illustrate how we can translate the insights from this model to real neural networks, we revisit the experiment from Figure 1. In this experiment we fit a neural network to predict two different sets of target values: first, we construct $y_i = \mathbf{z}(\mathbf{x}_i)^\mathsf{T} \mathbf{b}^*$, where $\mathbf{b}^*$ is chosen to provide the best linear fit of binary $\pm 1$ labels distinguishing between two classes of digits from the MNIST dataset using an independently initialized network of the same architecture. Therefore, $y_i$ are well represented by the initial tangent features, and by the final layer features $\mathbf{z}(\mathbf{x})$ in particular. Second, we use the

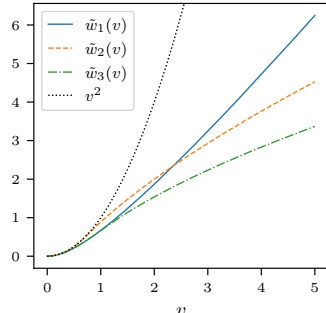
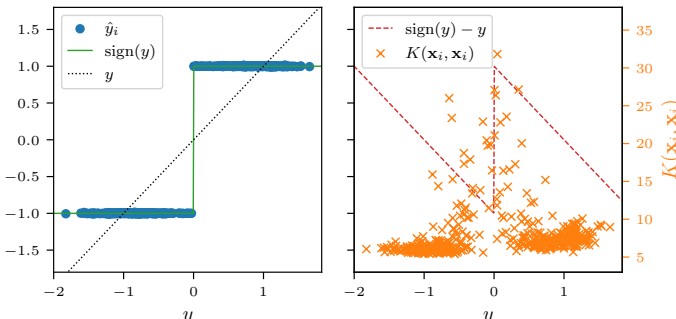

Figure 2: **Effective penalties are sub-quadrataic.** We plot effective penalties for $\omega_1(v) = (v-1)^2$, $\omega_2(v) = |v-1|$, and $\omega_3 = \omega_1 \oplus \omega_2$. All are sub-quadratic, yet all behave like $v^2$ near $v = 0$.

Figure 3: **Adapted kernel reveals difficult structure.** For the neural network from Figure 1 trained on binarized labels $\text{sign}(y_i)$ (**left**), the target function (green, solid) is difficult while the function $\mathbf{x} \mapsto y$ (black, dotted) is easily predicted using tangent features. The network must learn (**right**) to fit the residual (red, dashed), which results in the kernel (orange, $\times$) being highly influenced by difficult training points (near $y = 0$).

binarized labels $\text{sign}(y_i)$ (illustrated in Figure 3 (left)), which are not representable using only $\mathbf{z}(\mathbf{x})$ and therefore requires using the tangent features available at prior layers.

For the linear targets $y_i$, since only final layer features are required, the solution should have large values only in the final layer weights $\widehat{\mathbf{B}}_L$, which for this problem are simply a vector $\widehat{\boldsymbol{\beta}}_L$. Thus, we should expect only the final layer to contribute to the change in the adapted kernel:

$$K(\mathbf{x}, \mathbf{x}') \approx K_0(\mathbf{x}, \mathbf{x}') + ([\mathbf{s}_L^{(1)}]_1^2 - 1)\|\widehat{\boldsymbol{\beta}}_L\|_2^{-2}(\mathbf{z}(\mathbf{x})^\mathsf{T}\widehat{\boldsymbol{\beta}}_L)(\mathbf{z}(\mathbf{x}')^\mathsf{T}\widehat{\boldsymbol{\beta}}_L).$$

This final layer contribution should be very close to $K_{\widehat{\mathbf{y}}}(\mathbf{x}, \mathbf{x}')$, the label kernel. We can see that our theoretical adaptive feature model reflects real neural networks in Figure 1 (left), where the adapted kernel very closely resembles a linear combination of the initial NTK and the label kernel.

For the binarized targets $\text{sign}(y_i)$, which are poorly represented by the initial NTK features, the solution should require larger values of $\widehat{\mathbf{B}}_\ell$, which will result in more change to the adapted kernel. To understand where these large values will be allocated, consider that the linear targets $y_i$ are essentially the best linear fit of the binarized labels $\text{sign}(y_i)$, so the final layer contribution to the prediction should be proportional to $y_i$. Thus, the remainder of the weights only fit the residual $\text{sign}(y_i) - \alpha y_i$, which is largest for data points with $y_i$ near 0. The solution should allocate principal components that align with the tangent features at earlier layers of these data points, which function as "support vectors" of the solution, and the components should be larger as $y_i$ are nearer to 0. Indeed, this is exactly what we see in the real neural network in Figures 1 and 3 (right): the adapted kernel reflects the initial NTK, the label kernel for linear targets, and the label kernel for binary targets, but it is also very large for data points having $y_i$ near 0, where features aligning with points with large residuals must be amplified.

## 4. Discussion

Although our framework is far from capturing all of the complexities of neural networks, we have been able to shed some light on kernel alignment to close our gap in understanding. We now discuss a few possibilities for future work and connections to other closely related literature.

**Average features vs. final features.** In our analysis we considered the average features over a linear path in parameter space, but this is difficult to work with in practice, compared to for example the features at the end of training. Of course, we can equivalently write the average features as an

average transformation by $\mathbf{M}_{\boldsymbol{\theta}'}$ such that $\nabla_{\boldsymbol{\theta}'} f_{\boldsymbol{\theta}'}(\mathbf{x}) = \nabla_{\boldsymbol{\theta}_0} f_{\boldsymbol{\theta}_0}(\mathbf{x})\mathbf{M}_{\boldsymbol{\theta}'}$ along the path from $\boldsymbol{\theta}_0$ to $\boldsymbol{\theta}$:

$$\overline{\nabla f_{\boldsymbol{\theta}}}(\mathbf{x}) = \nabla_{\boldsymbol{\theta}_0} f_{\boldsymbol{\theta}_0}(\mathbf{x})\mathbf{M} = \nabla_{\boldsymbol{\theta}_0} f_{\boldsymbol{\theta}_0}(\mathbf{x})\Big(\mathbf{I}_P + \int_0^1 (\mathbf{M}_{\boldsymbol{\theta}'} - \mathbf{I}_P)\big|_{\boldsymbol{\theta}' = (1-t)\boldsymbol{\theta}_0 + t\boldsymbol{\theta}} dt\Big).$$

In general, an integral of matrices, even if individually low rank, should not be a low rank matrix. However, the average $\mathbf{M}$ is often a low-rank perturbation to $\mathbf{I}_P$, which is a remarkable coincidence unless all $\mathbf{M}_{\boldsymbol{\theta}'}$ along this path have the same principal subspace as $\mathbf{M}$ but with different eigenvalues. We thus expect the average features and final features to be similar, but not necessarily the same (in general, due to the averaging effect, the final tangent features will be larger for example). This difference can be important in downstream analysis—for example, in transfer learning, we would let $\boldsymbol{\theta}_0$ be the solution to a previous optimization problem, and the initial tangent features would be the final tangent features of that optimization, rather than the average tangent features. Thus another direction for further study is the extent to which average features and final features coincide. In our limited (unpublished) observations, the final feature kernel closely resembles a re-scaled version of the average feature kernel, which coincides with recent results for linear networks [19].

**Benign overfitting.** Neural networks have shown remarkable resilience to noisy labels, sparking the recent theoretical research area of studying models that interpolate noisy labels yet still generalize well [3, 9, 13, 31]. Much research in this area has been concerned with finding fixed feature regimes in which this "benign overfitting" can occur in ridge(less) regression settings, but Donhauser et al. [34] have shown that when the ground truth function is sparsely represented by the features, remarkably, the optimal $\ell_p$ penalty is not $p \in \{1, 2\}$, but rather in between, since some resemblance to sparsity-inducing $p = 1$ encourages learning structure, but something like $p = 2$ is necessary to absorb noise without harming prediction (for more discussion regarding the latter point, see [31]). Our results reflect these desired optimal properties precisely, since the effective penalties always have sub-quadratic tail behavior (promoting structure) and quadratic behavior near zero (absorbing noise). This raises another future research question, regarding the optimality of these effective penalties compared to $\ell_p$, $p \in (1, 2)$ penalties.

**Low rank optimization.** Our framework sheds interesting insight on the success of low rank deviation optimizations in neural networks, which have been used to characterize task difficulty and compress networks [37] and recently to efficiently fine-tune pretrained large language models in the low rank adaptation (LoRA) method [38]. In LoRA, for example, each weight matrix is parameterized as $\mathbf{W}_\ell = \mathbf{U}_\ell \mathbf{V}_\ell^\top$ for $\mathbf{U}_\ell \in \mathbb{R}^{P_\ell \times R_\ell}$ and $\mathbf{V}_\ell \in \mathbb{R}^{Q_\ell \times R_\ell}$, where $R_\ell \ll P_\ell, Q_\ell$. Due to the sub-quadratic spectral regularization of the adaptive feature optimization, if the target task is sufficiently related to the source task, such that the target function is well represented by the tangent features at $\boldsymbol{\theta}_0$ of the pretrained model, then according to Theorem 2, the model is inclined to learn an approximately low-rank deviation from the initial parameters even if the $\mathbf{W}_\ell$ were not explicitly constrained to be low rank. By constraining the weights to be low rank by design, LoRA can essentially recover the same solution, if not an even more aggressively structured one, at a fraction of the computational time and memory cost. An interesting question for future work is how close the solution using LoRA's hard low rank constraint is to the solution under the soft low-rank constraint of adaptive feature learning, and which of these solutions yields a better predictor.

**Effect of depth.** We speculate that the value of depth in a network is in providing a very rich set of late layer features from which a few meaningful principal components can be extracted in feature learning. The richer these features are, the easier it will be to fit a parsimonious model using only a few low rank components.

# Acknowledgements

The experiments in this paper were written in PyTorch [39] with code assistance from GitHub Copilot. DL was supported by ARO grant 2003514594. SA and computing resources were supported by Richard G. Baraniuk via NSF grants CCF-1911094, IIS-1838177, and IIS-1730574; ONR grants N00014-18-1-2571, N00014-20-1-2534, and MURI N00014-20-1-2787; AFOSR grant FA9550-22-1-0060; and a Vannevar Bush Faculty Fellowship, ONR grant N00014-18-1-2047.

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

# A. Theoretical technical details

## A.1. Diagonal $\mathbf{M}$ example

In the case that $\mathbf{M}$ is diagonal, the problem immediately separates into $P$ scalar optimizations of $\omega(m_j) + \theta_j^2 = \omega(m_j) + \frac{\beta_j^2}{m_j^2}$. Thus the effective penalty is a sum over these $P$ scalars:

$$\widetilde{\Omega}(\boldsymbol{\beta}) = \sum_{j=1}^{P} \tilde{\omega}(\beta_j).$$

Since $\tilde{\omega}$ is subquadratic, this is roughly a conventional sparsity inducing penalty, except since the behavior is quadratic near 0, it does not promote exact sparsity. If, however, $\omega(m) = m^2$ (which does not meet the requirements that we use elsewhere in this work, as it is not minimized at $\omega(1) = 0$), then the minimizer of each penalty is given by $m_j^2 = |\beta_j|$, resulting in an effective penalty of

$$\widetilde{\Omega}(\boldsymbol{\beta}) = \sum_{j=1}^{P} 2|\beta_j| = 2\|\boldsymbol{\beta}\|_1.$$

## A.2. Penalty function results

### A.2.1. Joint penalty quasi-convexity

**Proposition 4.** *If $\omega_1$ and $\omega_2$ are strictly quasi-convex functions minimized at $\omega_1(1) = 0$ and $\omega_2(1) = 0$, then $\omega_1 \oplus \omega_2$ is a strictly increasing function defined on $[1, \infty)$ such that $(\omega_1 \oplus \omega_2)(1) = 0$.*

*Proof.* Since we only evaluate $\omega_1$ and $\omega_2$ on $[1, \infty)$, the only important property is that they are both strictly increasing. It is clear that $(\omega_1 \oplus \omega_2)(1) = \omega_1(1) + \omega_2(1) = 0$. We need only show that the function is increasing.

First, we argue that the function $z \mapsto \omega_1(z) + \omega_2(\frac{v}{z})$ defined on $(0, \infty)$ for $v \geq 1$ always takes minimum value for $z \in [1, v]$. It can never be minimized for $z < 1$, since in that case $\omega_1(z) > \omega_1(1)$ and $\omega_2(\frac{v}{z}) > \omega_2(v)$. By symmetry, it can also not be minimized for $z > v$. Now let $1 \leq v_1 < v_2$. Then

$$(\omega_1 \oplus \omega_2)(v_2) > \min_{1 \leq z \leq v_2} \omega_1(z) + \omega_2(\tfrac{v_1}{z}) = \min_{1 \leq z \leq v_1} \omega_1(z) + \omega_2(\tfrac{v_1}{z}) = (\omega_1 \oplus \omega_2)(v_1),$$

and therefore $\omega_1 \oplus \omega_2$ is strictly increasing on $[1, \infty)$. $\qquad\square$

### A.2.2. Proof of Proposition 3

*Proof.* To see that $v^2 \mapsto \tilde{\omega}(v)$ is concave, note that $v^2 \mapsto -\tilde{\omega}(v)$ is the Legendre–Fenchel conjugate of the function $z^2 \mapsto \omega(\frac{1}{z^2})$, which is therefore convex. The proof that $\tilde{\omega}$ is increasing is similar to the proof of Proposition 4: the optimal $z^*$ cannot be less than 1, and note that for all $z \geq 1$, if $v_1 < v_2$,

then $\frac{v_1^2}{z_1^2} < \frac{v_2^2}{z_2^2}$, and therefore $\tilde{\omega}(v_1) < \tilde{\omega}(v_2)$, since the minimizer must be finite. Lastly, we obtain the second-order Taylor series expansion about $v = 0$. Note that at $v = 0$, $z^* = 1$. Then

$$\partial \omega(z^*) - 2\frac{v^2}{z^{*3}} \ni 0 \implies \left.\frac{\partial \tilde{\omega}(v)}{\partial v}\right|_{v=0} = \left.2\frac{v}{z^{*2}}\right|_{v=0} = 0$$

$$\text{and} \implies \left.\frac{\partial^2 \tilde{\omega}(v)}{\partial v^2}\right|_{v=0} = 2\frac{1}{z^{*2}} - 4\frac{v}{z^{*3}}\left.\frac{\partial z^*}{\partial v}\right|_{v=0} = 2.$$

To justify the latter, consider two cases: if $\omega$ is non-differentiable at $z = 1$, then for sufficiently small $v$, $z^* = 1$ must be constant as $2v^2 \in \partial \omega(z^*)$; otherwise, for some $p \geq 2$, $\omega(z) = |z - 1|^p + o(|z - 1|^p)$. In the second case, for small $v$, this means that $p|z^* - 1|^{p-1} \approx 2v^2$ and this approximation becomes exact as $v \to 0^+$, so taking the limit of the derivative and plugging in $p|z^* - 1|^{p-1} = 2v^2$,

$$p(p-1)|z^* - 1|^{p-2}\frac{\partial z^*}{\partial v} = 4v \implies \frac{\partial z^*}{\partial v} = C_p v^{1 - \frac{2(p-2)}{p-1}}.$$

This implies that $v\frac{\partial z^*}{\partial v} = C_p v^{\frac{2}{p-1}} \to 0$ as $v \to 0^+$. As a result, we have the Taylor expansion $\tilde{\omega}(v) = 0 + (0)v + \frac{(2)}{2}v^2 + o(v^2)$, as stated. $\qquad\square$

## A.3. Proof of main results

To prove the main theorems, we first prove the following lemma.

**Lemma 5.** *Given strictly quasi-convex $\omega_1$ and $\omega_2$ minimized at $\omega_1(1) = 0$ and $\omega_2(1) = 0$ and a constraint set $\mathcal{B} \subseteq \mathbb{R}^{P \times Q}$, there is a solution*

$$\widehat{\mathbf{M}}_1, \widehat{\mathbf{M}}_2, \widehat{\mathbf{W}} \in \underset{\substack{\mathbf{M}_1 \in \mathbb{R}^{P \times P}, \\ \mathbf{M}_2 \in \mathbb{R}^{Q \times Q}, \\ \mathbf{W} \in \mathbb{R}^{P \times Q}}}{\arg\min} \Omega_{\omega_1}(\mathbf{M}_1) + \Omega_{\omega_2}(\mathbf{M}_2) + \|\mathbf{W}\|_F^2 \; \text{s.t.} \; \mathbf{M}_1 \mathbf{W} \mathbf{M}_2 \in \mathcal{B}$$

*having the form $\widehat{\mathbf{M}}_1 = \mathbf{U}\mathbf{S}_1\mathbf{U}^\top$, $\widehat{\mathbf{M}}_2 = \mathbf{V}\mathbf{S}_2\mathbf{V}^\top$, $\widehat{\mathbf{W}} = \mathbf{U}\boldsymbol{\Sigma}\mathbf{V}^\top$, where $\mathbf{S}_1 = \text{diag}(\mathbf{s}_1)$, $\mathbf{S}_2 = \text{diag}(\mathbf{s}_2)$, $\boldsymbol{\Sigma} = \text{diag}_{P \times Q}(\boldsymbol{\sigma})$, and $\mathbf{D} = \text{diag}_{P \times Q}(\mathbf{d})$ such that for $j \leq \min\{P, Q\}$*

$$[\mathbf{s}_1]_j, [\mathbf{s}_2]_j, [\boldsymbol{\sigma}]_j = \underset{s_1, s_2 \geq 1, \sigma \geq 0}{\arg\min} \omega_1(s_1) + \omega_2(s_2) + \sigma^2 \; \text{s.t.} \; s_1 s_2 \sigma = [\mathbf{d}]_j \qquad (3)$$

*and $[\mathbf{s}_1]_j = 1$, $[\mathbf{s}_2]_j = 1$ for $j > \min\{P, Q\}$, and*

$$\widehat{\mathbf{B}} = \mathbf{U}\mathbf{D}\mathbf{V}^\top \in \arg\min_{\mathbf{B} \in \mathcal{B}} \widetilde{\Omega}_{\omega_1 \oplus \omega_2}(\mathbf{B}).$$

*Proof.* First we show that the singular vectors are shared. Fix a matrix in the constraint set $\mathbf{B} \in \mathcal{B}$. Define the singular value decompositions $\mathbf{M}_1 = \mathbf{U}\mathbf{S}_1\mathbf{U}^\top$ and $\mathbf{M}_2 = \mathbf{V}\mathbf{S}_2\mathbf{V}^\top$, and let $\boldsymbol{\Sigma} = \mathbf{U}^\top\mathbf{W}\mathbf{V}$. Then we have the linear constraint $\mathbf{U}^\top\mathbf{B}\mathbf{V} = \mathbf{S}_1\boldsymbol{\Sigma}\mathbf{S}_2$, or equivalently $\boldsymbol{\Sigma} = \mathbf{S}_1^{-1}\mathbf{U}^\top\mathbf{B}\mathbf{V}\mathbf{S}_2^{-1}$. Thus $\|\mathbf{W}\|_F = \|\boldsymbol{\Sigma}\|_F$ is minimized when $\mathbf{U}$ and $\mathbf{V}$ are chosen to align the smallest values of $\mathbf{S}_1^{-1}$ and $\mathbf{S}_1^{-2}$ with the largest singular values of $\mathbf{B}$—in other words, $\mathbf{U}$ and $\mathbf{V}$ are the left and right singular vectors of $\mathbf{B}$, respectively. We thus have the singular value decompositions $\mathbf{W} = \mathbf{U}\boldsymbol{\Sigma}\mathbf{V}^\top$ and $\mathbf{B} = \mathbf{U}\mathbf{D}\mathbf{V}^\top$ where $\mathbf{D} = \mathbf{S}_1\boldsymbol{\Sigma}\mathbf{S}_2$.

It is now clear that the optimization is a sum over aligned eigenvalues and so is equivalent to eq. (3). Any values of $\mathbf{s}_1$ or $\mathbf{s}_2$ not part of this optimization $(j > \min\{P, Q\})$ must be 1. Now observe that

$$\underset{s_1, s_2 \geq 0}{\text{minimize}} \; \omega_1(s_1) + \omega_2(s_2) \; \text{s.t.} \; s_1 s_2 = a$$

is equivalent to solving

$$\underset{s_1, s_2 \geq 0}{\text{minimize}} \; \omega_1(s_1) + \omega_2\left(\frac{a}{s_1}\right)$$

and taking $s_2 = \frac{a}{s_1}$, but this is simply the definition of $\omega_1 \oplus \omega_2$. By Proposition 4, we must have $s_1, s_2 \geq 1$, and then by the definition of the effective penalty, we obtain the optimization over $\mathbf{B}$. $\qquad\square$

### A.3.1. Proof of Theorem 1

*Proof.* We apply Lemma 5 with $Q = 1$, $\mathbf{W} = \boldsymbol{\theta} - \boldsymbol{\theta}_0$, $\mathcal{B}$ equal to the linear constraint set, $\omega_1 = \omega$, and $\omega_2 = \chi_{\{1\}}$, forcing $\mathbf{M}_2 = 1$, so we have only $\mathbf{M}_1 = \mathbf{M}$. Now $\omega_1 \oplus \omega_2 = \omega_1 = \omega$. $\widehat{\mathbf{M}}$ must have all but one eigenvalues equal to 1, and the other eigenvalue $s$ must have corresponding eigenvector aligned with $\widehat{\mathbf{B}} = \widehat{\boldsymbol{\beta}}$. Note that the singular value of $\mathbf{W}$ is simply $\sigma = \|\widehat{\boldsymbol{\theta}} - \boldsymbol{\theta}\|_2$, similarly for $\mathbf{B}$ the singular value is $d = \|\widehat{\boldsymbol{\beta}}\|_2$. We must have $s\sigma = d$, which gives the form of the stated results.

For the adapted kernel, we simply evaluate

$$\nabla_{\boldsymbol{\theta}_0} f_{\boldsymbol{\theta}_0}(\mathbf{x}) \widehat{\mathbf{M}}^2 \nabla_{\boldsymbol{\theta}_0} f_{\boldsymbol{\theta}_0}(\mathbf{x}')^\mathsf{T} = \nabla_{\boldsymbol{\theta}_0} f_{\boldsymbol{\theta}_0}(\mathbf{x}) \nabla_{\boldsymbol{\theta}_0} f_{\boldsymbol{\theta}_0}(\mathbf{x}')^\mathsf{T} + (s^2 - 1) \nabla_{\boldsymbol{\theta}_0} f_{\boldsymbol{\theta}_0}(\mathbf{x}) \|\widehat{\boldsymbol{\beta}}\|_2^{-2} \widehat{\boldsymbol{\beta}} \widehat{\boldsymbol{\beta}}^\mathsf{T} \nabla_{\boldsymbol{\theta}_0} f_{\boldsymbol{\theta}_0}(\mathbf{x}')^\mathsf{T},$$

which is equal to the stated adapted kernel. $\qquad\qquad\square$

### A.3.2. Proof of Theorem 2

*Proof.* We apply Lemma 5 for each $\ell$. The only care needed is in defining the constraint set $\mathcal{B}$. Proceeding in $\ell$ and fixing choices of the previous $(\mathbf{B}_{\ell'})_{\ell'=1}^{\ell-1}$, we can always define $\mathcal{B}$ in terms of the remaining $\mathbf{B}_\ell$ that can satisfy the linear constraint. For all such paths, the resulting equivalent optimization has the same form, and so we have the stated equivalent optimization for the linear constraint involving all $(\mathbf{B}_\ell)_{\ell=1}^L$. $\qquad\qquad\square$

## B. Experimental details

Code is available at `https://github.com/dlej/adaptive-feature-perspective`.

### B.1. MNIST regression experiment

In this experiment we use a 3 layer multi-layer perceptron (MLP) with 128 units at each layer and ReLU activation implemented in PyTorch [39] and with no bias. The network had a single output for regression. We used $N = 500$ training points from digits 2 and 3 from MNIST. We vectorized each image and normalized it to have zero mean and unit variance. For constructing target labels $y_i$, we froze the first two layers of a random Gaussain initialization of the network and only trained the last layer on predicting $\pm 1$ labels corresponding to classes 2 and 3 with a best linear fit. The resulting $y_i$ had 95.2% accuracy when thresholded at 0 in predicting the original classes. Then we trained two networks newly initialized networks, one to predict $y_i$, and the other to predict $\mathrm{sign}(y_i)$. For training we used stochastic gradient descent (SGD) with initial learning rate of 0.1 and a cosine annealing schedule, and 0.9 momentum.

To plot the kernels, we compute the average tangent features evaluated at 50 points along the linear path from $\theta_0$ to $\theta$. We order the points according to increasing value of $y_i$. We use different color scales for each kernel: for $K_0$, we use a maximum value of 4, for $K$, we use a maximum value of 10, and for $K_{\hat{y}}$, we use a maximum value of 2.

