# OpenReview forum: "An Adaptive Tangent Feature Perspective of Neural Networks"
_CPAL.cc/2024/Conference — CPAL 2024 (Proceedings Track) Oral_

### Official Review · Reviewer_5sab · 2023-10-02
**Report on An Adaptive Tangent Feature Perspective of Neural Networks**

**Rating:** 6
**Confidence:** 3

**Review:**

In this paper, the authors explore the application of linear transformations to features, leading to a joint optimization encompassing parameters and transformations, while adhering to a bilinear interpolation constraint. Additionally, they demonstrate how constraining adaptivity imparts specific regularization characteristics to the solution, resulting in a group approximate low-rank penalty on a neural network-based model. This provides a framework for understanding the properties of neural networks. In summary, this paper presents an interesting and valuable contribution.

Comments:

1. In Line 146, you assume that $M$ is symmetric positive semidefinite, but it would be helpful to explain how this condition is guaranteed through the rotation of $\theta$. Furthermore, I am curious if your conclusions are applicable to symmetric matrices, which are more commonly encountered in practical implementations.

2. Reviewers have suggested that the authors provide a simplified proof sketch in main paper to assist readers in grasping the key points of the proof more easily. This addition would enhance the paper's accessibility and comprehension.

---

### Official Review · Reviewer_45h3 · 2023-10-06
**Review of Submission 32: An Adaptive Tangent Feature Perspective of Neural Networks**

**Rating:** 6
**Confidence:** 2

**Review:**

## Overview:

In this paper, the authors "propose a framework for understanding linear models in tangent feature space where the features are transformed". This enables them to gain new insight on settings that have challenged the traditional neural tangent kernel analysis, particularly in realistic, finite size neural networks that are trained on standard data (e.g., MNIST, CIFAR-10).

## Strengths:

1. The paper deals with an interesting and important goal of extending analysis beyond the NTK, where the gradients are fixed.
2. The authors develop a general framework, which enables them to study a range of problems.
3. The authors are upfront with their models' assumptions and limitations. They should be applauded for their honesty and directness.
4. The analysis provides enhanced (by an *order* of magnitude) low-sample performance of feature learning, as compared to the traditional fixed gradient analysis.

## Weaknesses:

In general, I found this paper challenging to follow. I think this is mostly a failing on my end (hence, my low confidence), but I do think there are several ways in which the authors could make their work more clear.

1. Low-rankedness: How do the assumptions shape the identified "low-rank perturbation" in Theorem 1? Is it a direct consequence of the assumptions, or a more general aspect of the adaptive feature perspective? I am also unclear whether the low-rank perturbation necessarily counts as a low-rank solution, as the phrasing in the abstract suggests ("with structured regularization that encourages approximately low rank solutions" [lines 6-7]).
2. A little more detail on how, in the structureless feature learning case (where features are able to evolve) the same solution as the standard NTK analysis emerges, would be helpful. From the motivation of the work in the introduction, I found this surprising, as I was under the assumption that changes in features would necessarily lead to a departure from the NTK.
3. More discussion surrounding the results of Theorem 2 would also be helpful. I found them challenging to interpret, and the numerical examples, while seemingly promising, were not explained in as much detail as I needed to fully understand them. Particularly, I did not feel like I fully understood Figure 3.

## Summary:

This paper tackles an interesting and important question, which is relevant and aligned CPAL's research areas.  While the numerical results seemed promising, I did not feel like I fully understood them, nor the analytical theory. If the authors could provide some additional insight, I would very much consider increasing my score.

---

### Official Review · Reviewer_L6Tc · 2023-10-07
**An Adaptive Tangent Feature Perspective of Neural Networks**

**Rating:** 5
**Confidence:** 2

**Review:**

This paper studies linear models in tangent feature space with transformable features. This work indicates the relationship between linear feature adaptivity and structured regression using fixed features with low-rank constraints. The authors provide insights into how the features and kernel functions change. Experiments support the results. My concerns are the following.

1. This paper is not very easy to follow. The writing needs improvement.

2. There are too many assumptions about the Adaptive feature model and the neural network model. I feel it makes the conclusion weak.

3. The practical significance of the results is not very clear. For example, if the authors want to relate the results with LoRA, it is better to provide a Corollary with a discussion on LoRA.

----------------------------------------------------------------------------------------------------------------------------
Thank you for the reply, and sorry for the late reply. I am partially satisfied with the response. For the examples, I am referring to whether your assumption applies to some common neural networks, such as convolutional networks with ReLU activations. It is difficult to make a decision. I will keep my rate of 5 but decrease my confidence to 2.

---

### Meta-Review · Area_Chair_v4PM · 2023-11-09

**Recommendation:** Accept (Poster)
**Confidence:** 4

**Metareview:**

This paper proposes an extension of neural tangent kernel framework where the tangent features are allowed to adapt to the
data along with the regression coefficients. In particular, the authors propose jointly optimizing for a linear transformation of the tangent features and regression coefficients under an interpolation constraint. This is shown to be equivalent to a linearly constrained optimization problem with structured regularization that encourages approximately low rank solutions.

Reviews are mixed but skewing positive (5,6,7). Reviewers praised the work for analyzing a regime that goes beyond the NTK, and its insights into how the features and kernel functions may change when training neural networks. However, the clarity and accessibility of the paper seem to be a common concern among reviewers. The validity of the results was not called into question, but several reviewers thought more discussion surrounding the main theorems would be helpful, as well as more details for the numerical experiments. This was addressed directly by the authors in the rebuttal to the satisfaction of the reviewers.

Overall, despite some issues with clarity/presentation, the consensus is that this has paper important insights for training outside NTK regime and will be of interest to the CPAL audience.

---

### Decision · Program_Chairs · 2023-11-19

**Decision:**

Accept (Oral)

**Comment:**

The paper aims to go beyond the NTK regime by allowing the tangent features to adapt to the data. They suggested that the optimization problem can be reformulated and lead to a structural regularization that promotes low-rank solutions. The reviews are somewhat mixed, where there is an appreciation for going beyond the NTK regime and yet, the paper's clarity has been called into questions to hinder understanding. The latter issues have been partially addressed during the rebuttal, and it is recommended that the authors further polish the writing, and include more discussions around the results.

The action PC chair for this paper is Yuejie Chi, who made the decision after carefully reading the paper as well as the comments by all reviewers and AC. The decision is agreed by all PC chairs.